# Outcomes of Laryngeal Cancer Surgery after Open Partial Horizontal Laryngectomies with Lateral Cervical Approach

**DOI:** 10.3390/jcm11164741

**Published:** 2022-08-14

**Authors:** Andrea Colizza, Massimo Ralli, Arianna Di Stadio, Francesca Cambria, Federica Zoccali, Fabrizio Cialente, Diletta Angeletti, Antonio Greco, Marco de Vincentiis

**Affiliations:** 1Department of Sense Organs, Sapienza University of Rome, 00161 Roma, Italy; 2Department GF Ingrassia, University of Catania, 95125 Catania, Italy

**Keywords:** open partial horizontal laryngectomies, lateral cervical approach, laryngeal squamous cell carcinoma, penetration aspiration scale, anterior myocutaneous flap

## Abstract

Background: Open partial horizontal laryngectomies (OPHL) are one of the surgical techniques used for the conservative management of laryngeal cancers. The aims of this study are to analyze the oncological and functional results of a group of patients affected by laryngeal squamous cell carcinoma (LSCC) treated with OPHL, performed using a minimally invasive technique. Methods: This is a prospective case–control study. We enrolled 17 consecutive patients with LSCC treated with OPHL through a lateral cervical approach (LCA). Patients were evaluated using their Penetration Aspiration Scale score (liquid, semiliquid and solid) and Voice Handicap Index (VHI) at three different endpoints: 15 days (T1), 3 months (T2), and 6 months (T2) after surgery. Results: The functional outcomes of the LCA are stackable with that of the classical anterior cervical approach in terms of respiration, swallowing, and speech. One-way ANOVA was performed to evaluate the variances of PAS and VHI scores at the three different observation points. No statistically significant differences were observed between OPHL- PAS scores for liquid (*p* = 0.1) at the three different observation points. A statistically significant improvement was observed in the OPHL- PAS score for semisolids and solids (*p* < 0.00001) between T1 and T3 (*p* = 0.0001) and for solids between T2 and T3 (*p* < 0.00001). The improvement of VHI-10 was statistically significative (*p* < 0.00001) at the three different observation points (T1–T2 and T2–T3). Conclusion: The LCA is a potential approach for laryngeal surgery in selected cases. The preoperative staging and planning are of the utmost importance to ensure oncological radicality. The main advantage of this approach is the preservation of the healthy tissues surrounding the larynx and the functional and oncological outcomes are stackable with the classic anterior cervical approach.

## 1. Introduction

Open partial horizontal laryngectomy (OPHL) is one of the most common surgical techniques in the conservative management of laryngeal cancers. The concept of partial and conservative laryngeal surgery was first introduced nearly 70 years ago by Alonso [1] and subsequently developed as a new surgical technique [2,3,4]. In the last few years, minimally invasive techniques for partial laryngectomy, total laryngectomy, and salvage laryngo-pharyngectomy after chemo-radiotherapy have been described [4,5,6]. These mini-invasive approaches aim to preserve, as much as possible, the healthy tissues surrounding the larynx. Recently, a lateral cervical approach (LCA) has been described; it allows neck dissection and open laryngectomy whilst respecting the oncological radicality and preserving the surrounding healthy tissues at the same time [7,8,9]. This approach is performed by making a skin incision following the anterior border of the sternocleidomastoid muscle, followed by the preparation of an anterior myocutaneous (AMC) flap in order to create a space to perform the lateral neck dissection, the partial laryngectomy, and the pexy.

The purpose of this prospective case–control study is to describe the oncological and functional outcomes in patients affected by laryngeal squamous cell carcinoma (LSCC) undergoing OPHL and neck dissection through LCA.

## 2. Materials and Methods

### 2.1. Study Cohort

Between September 2019 and April 2021, we performed OPHL and neck dissection with minimally invasive LCA on 17 patients with a diagnosis of LSCC at the Department of Otorhinolaryngology of our University Hospital [BLINDED]. This study was conducted according to the principles of the Helsinki Declaration and in accordance with the regulations of the Ethical Committee for retrospective studies of the involved institution. Informed consent was obtained from all the participants before their inclusion in the study.

Inclusion and exclusion criteria were defined.

Inclusion criteria were: (i) anatomopathological diagnosis of LSCC performed through previous biopsy and (ii) clinical TNM stage amenable to perform an OPHL according to NCCN guidelines.

Exclusion criteria were: (i) previous radiotherapy in the head and neck region, other histological types of cancers of the larynx, and (ii) cTNM stage not suitable for OPHL surgery. No other exclusion criteria were applied. According to the National Comprehensive Cancer Network (NCCN) 2022 guidelines, conservative approaches, such as endoscopic resection or open partial resections, are recommended clinically for T1, T2, or selected T3 tumors.

All patients preoperatively underwent otolaryngology examination and narrowband imaging (NBI) endoscopy. Microlaryngoscopy with biopsy of the laryngeal lesion confirmed the diagnosis of LSCC, and a total body Computed Tomography (CT) scan was performed to stage the lesion. Clinical information including age, gender, smoking history, and comorbidities was collected at the first visit for each patient.

The clinical and pathological staging was calculated according to the Eight Edition of the AJCC Cancer Staging Manual.

### 2.2. Surgical Technique

Surgery was performed by the same surgeon. Neck dissection through LCA was performed according to international guidelines; in detail, a monolateral or bilateral linear incision following the anterior border of the sternocleidomastoid muscle was performed to harvest an anterior myocutaneous (AMC) flap. This flap was elevated from the lower muscles (sternothyroid and thyrohyoid muscles) in order to create a space of work between the omohyoid muscle to one side and sternohyoid and thyrohyoid muscles to other side. Through this approach, it was possible to perform the lateral neck dissection, the OPHL, and the pexy. Tracheostomy was performed through a second skin incision separated from the lateral one.

In Figure 1A, there is a preoperative CT scan with a contrast agent. This scan shows a laryngeal tumor of the left vocal folds. Figure 1B shows the same patient six months after surgery with a new larynx without the thyroid cartilage.

### 2.3. Functional Evaluation

Respiration was evaluated with the decannulation rate and the mean time of decannulation after surgery.

Swallowing was evaluated through Fiberoptic Endoscopic Evaluation of Swallowing (FEES) [10]. Endoscopic evaluation was conducted by the same physician (A.C.) using a flexible endoscope connected to a camera and a high-definition monitor (Full HD). The exam was recorded and independently evaluated some days later by two other researchers (M.R. and F.C.) to confirm the results. Swallowing of liquids, semisolids, and solids was assessed using blue dyed water, pudding, and crackers. A 5cc bolus was given to each participant three times for liquids and semisolids. Three trials with a quarter of an 8g cracker were carried out for solids. Penetration or aspiration in the neolarynx and the cough ability was evaluated through the Penetration Aspiration Scale adapted to open partial laryngectomy (OPHL-PAS) according to Pizzorni et al. OPHL-PAS is an 8-point tool adapted to the anatomy of the neolarynx [11]. This score is categorized in OPHL-PAS for OPHL type 1, OPHL type IIa-IIIa and OPHL type IIb-IIIb. These evaluations were repeated 15 days, 3 months, and 6 months after surgery.

Phonation was evaluated with the short version of the Voice Handicap Index (VHI-10) validated in Italian by Forti et al. [12]. The VHI is an instrument designed to examine the self-perceived emotional, physical, and functional effects of patients’ voice dysfunction [13]. The VHI-10 is a valid instrument, with a slight loss of information, with respect to the VHI-30, to quantify the patients’ own perceptions of voice deficiencies in less time than the VHI-30 [13]. The higher the VHI-10 score, the greater the problem of the voice. These evaluations were repeated 15 days, 3 months, and 6 months after surgery.

### 2.4. Statistical Analysis

All statistical analyses were performed using SPSS Version 25.0 (IBM Corp, Armonk, NY, USA). Descriptive analyses were mainly applied. Data are indicated as mean ± standard deviation, range, and percentage. One-way ANOVA with Bonferroni Holmes (BH) test was used to evaluate the difference in the score of OPHL-PAS, liquid, semisolid, and solid separately between T1, T2, and T3. The same was carried out to analyze VHI scores. A value of *p* < 0.05 was considered statistically significant.

## 3. Results

### 3.1. Demographic Features

In this study, we enrolled 15 males and 2 females. The mean age was 63.8 years (range 57–79), like data available in the literature [14,15]. Dysphonia was the most common symptom at presentation and only three patients had associated dysphagia. None of them had undergone radiotherapy or chemotherapy treatment before the surgery. 15 patients were cN0, one patient was cN2a and one was cN1. All patients were clinically M0. The main demographics, surgical features, and oncological outcomes of patients are summarized in Table 1.

### 3.2. Oncological Outcomes

After a multidisciplinary discussion between surgeons, oncologists, and radiotherapists, patients underwent surgical treatment.

Each patient underwent OPHL performed with the LCA approach. In particular, three patients underwent OPHL 1, six patients OPHL 2A, and eight patients underwent OPHL 2B according to the classification proposed by the European Laryngological Society (ELS) [16]. Eleven patients underwent ipsilateral selective neck dissection, and six patients underwent bilateral selective neck dissection [17].

Neck lymph node metastases were found in four cases (case 1 was pN2a, case 2 was pN1, case 11 was pN2b, and case 16 was pN2a). None of the patients presented post-operative complications, such as bleeding or infection.

The mean surgical time was 199 min (range: 120–285 min). Seven patients performed adjuvant radiotherapy after surgery (cases 1, 3, 6, 14, 11, 16, and 17); one patient refused the therapy (case 3).

Follow-up time was 17.4 (mean) months (range: 9–27). One patient died due to the disease (DOD) (case 3), another, still alive, had a recurrence of the disease after 16 months (AWD) (case 5) and two patients died due to other causes (DOC) (cases 8 and 14). The remaining 14 patients are alive without evidence of disease recurrence (NED).

### 3.3. Functional Outcomes

Respiration: Among the enrolled patients, 16 were decannulated after six months of observation (end of our study) (Table 2). The tracheostomy tube was removed 12.9 ± 9.9 days post-surgery (range 5–37). Among the decannulated patients,14 were decannulated at hospital discharge and two were decannulated during the post-operative period (cases 14 and 16). In only one case was it not possible to remove the tracheotomy tube during the six months after surgery (patient 11).

Among the sixteen patients decannulated, two of them (patients 5 and 12) required a second tracheostomy 17 and 29 days, respectively, after discharging due to oedema of the neolarynx and subsequent acute respiratory failure.

Swallowing: The oral feeding during the rehabilitation protocol started 11.9 ± 4.9 days after surgery. The NGT was removed 52.6 ± 37.2 days post-operatively (range: 24–183). After 3 months from the beginning of the rehabilitation program, only one patient (case 3) still had NGT due to a reduction in the sensory function on the neolarynx and underwent percutaneous endoscopic gastrostomy (PEG). At the end of six months of observation, it was possible to remove the nasogastric tube (NGT) in 16/17 patients (Table 2).

FEES evaluation was repeated 15 days, 3, and 6 months after surgery (Table 3). No statistically significant improvements were observed in the OPHL-PAS score for liquid (ANOVA: *p* = 0.1) at the three different observation points (Figure 2A). For the OPHL-PAS score for semisolid bolus, a statistically significant improvement was observed over the six months of observation (ANOVA: *p* = 0.0003); in particular, a significant difference was observed between T1 and T3 (BH: *p* = 0.0001) (Figure 2B). A statistically significant difference was observed for the OPHL-PAS score for solid over the period of the study (ANOVA: *p* < 0.00001); in particular, a significant difference improvement was observed between T1 and T3 (BH: *p* = 0.0001) and T2 and T3 (HB: *p* < 0.00001) (Figure 2C).

Phonation: The VHI-10 questionnaire was administered 15 days, 3, and 6 months after surgery. The mean VHI-10 at 15 days post-operatively was 17.5 ± 2.8, at 3 months was 15.9 ± 3.1, and six months after surgery was 12.9 ± 2.36. Table 4 details the values for each patient. Figure 2 shows the trend of VHI-10 values.

The variation of VHI-10 among the three observation points was statistically significant (ANOVA: *p* < 0.00001). A statistically significant improvement was observed between T1 and T3 (BH: *p* < 0.00001) and T2 and T3 (BH: *p* = 0.002) (Figure 3).

We analyzed the functional outcomes according to the surgical procedure. In Table 5, we divided the patients into OPHL I, OPHL IIA, and OPHL IIB and we compared the mean values and standard deviations of each procedure 6 months after surgery.

## 4. Discussion

Conservative laryngeal surgery is a useful tool for managing laryngeal cancer. In selected patients, according to the NCCN treatment guidelines, this surgery allows oncological radicality and the preservation of the main laryngeal functions.

The preservation of the larynx is important to preserve the functions of this organ and to speed and optimize patients’ recovery after surgery. This is the concept behind the development of various conservative surgical techniques such as OPHL. By means of LCA, the concept of preserving healthy tissues is extended to extralaryngeal structures. Spriano et al. [7] first described this surgical procedure and underlined that the possibility to perform the lateral neck dissection at the same time as the laryngeal surgery was the best advantage of this technique. In the literature, transoral laser approaches for locally advanced (T3 and T4a) supraglottic cancers have been described [18,19]. Nevertheless, with this transoral approach, the lateral neck dissection is performed with an open surgery approach, making a skin incision like the LCA one.

Laryngeal conservative surgery aims to achieve oncological radicality and preserve laryngeal functions. The full recovery of autonomous breathing is the first parameter for evaluating the success of OPHL; decannulation rate and the post-operative day of decannulation are two important findings to evaluate. Patients’ decannulation likelihood after OPHL ranges between 93% and 99% [20,21,22], but it is important to underline that this percentage varies among the various types of partial laryngectomy. Previous works reported different decannulation rates of 85.7% (134/149 patients) and 87% (33/38 patients) after cricohyoidopexy (OPHL 2B), and 98% (50/51 patients) after cricohyoidoepiglottopexy (OPHL 2A) [23,24]. We had only one patient who had to maintain the cannula after OPHL 2B.

Different decannulation times have been described from 8 days [20] to 91 days post-surgery [25]. Early decannulation after OPHL is still debated; some authors affirm that early decannulation can preserve from penetration and aspiration of secretions in the airways and allow better and faster recovery of swallowing. Looking at our results, an early decannulation (as soon as possible) during the rehabilitation protocol improves the sensitivity of the neolarynx during airflow and the laryngeal vibratory arrangement, and prevents the stiffness of the arytenoid [10]. Moreover, the tracheostomy decreased laryngeal elevation, desensitized the cough reflex, and caused less laryngeal closure [19]. These data are supported by Bron et al. [26]—who observed that early decannulation and tracheostomy closure are important to promote full swallowing recovery, prevent the ankylosis of the cricoarytenoid unit, stimulate a satisfactory cough reflex, and allow a more efficient closure of the neolarynx sphincter.

The recovery of the swallowing function is another important parameter after OPHL. Oral intake has been reported to range from 3 to 45 days after surgery [27]. In our study, we identified 11.9 mean post-operative days of restarting oral feeding, consistent with other studies. The removal time of NGT is still controversial, with a range between 3 and 107 days [28]. A previous multicentric retrospective series reported 25 days as the median removal time of NGT; these patients underwent partial laryngeal surgery after radiotherapy and chemotherapy treatments [29], so this was different from our study in which we included naïve patients, who were subjected to adjuvant treatments only after surgery.

The evaluation of swallowing is fundamental in the FEES protocol. We observed 4.5, 3.3, and 2.2 for the three consistencies of bolus as mean OPHL-PAS scores six months after surgery. The literature on this topic is scarce. Schindler et al. [30] recruited patients from 7 to 94 months after supratracheal laryngectomy (OPHL 3) and reported 4.1, 1.7, and 3.1 mean PAS scores, respectively, for liquid, semisolid, and solid bolus. The same authors in a different study [31] reported PAS scores after OPHL 2A of 6, 1, and 2 for liquids, semisolids, and solids.

We need a voice to maintain relationships and for our personal and social lives. Thus, every surgery which may cause a voice modification impacts the patient’s life. However, the handicap due to voice modification is generally secondary to oncological results [32]. After OPHL, in particular, types II and III, the neolarynx lacks vocal cords [33], so the capacity of preserving the voice is deputed to the mobility of the residual cricoarytenoid unit and patency of supraglottic tissue; the latter creates the neo-glottal sphincter complex that vibrates during phonation [34]. Thus, the sound generated in the neo-glottal sphincter is transmitted by vibration to above irregular mucosal structures; this explains the substantial roughness and breathiness of voice [35]. The disability due to voice alteration has been widely analyzed in the literature [30,34,36,37,38], but it is scarce if we focus on functional results (VHI) after OPHL. To the best of our knowledge, only Miyamaru and colleagues [38] analyzed voice dysfunctions after OPHL using the VHI-10. Our data showed an improvement in the VHI-10 score, which was statistically significant over time. Based on this preliminary finding, OPHL surgery through LCA might improve voice quality similarly to the classic approach.

### Limits of the Study

The main limit of this study is the small sample size; further studies with larger samples are necessary to confirm our preliminary findings.

## 5. Conclusions

LCA is a minimally invasive approach that can be used with successful results in selected patients. Correct preoperative staging and planning are mandatory to ensure oncological radicality. This technique allows the preservation of the healthy tissue surrounding the larynx and the functional outcomes are stackable with the ones of the classical anterior cervical approach in terms of respiration, swallowing, and speech. Further studies on larger samples are necessary to confirm our findings in terms of functional and oncological outcomes.

## Figures and Tables

**Figure 1 jcm-11-04741-f001:**
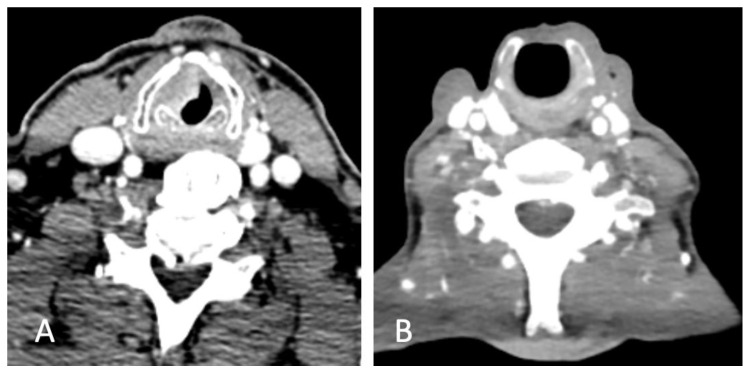
(**A**) CT scan of laryngeal carcinoma of the left vocal fold. (**B**) Post-operative CT scan of new laryngeal anatomy.

**Figure 2 jcm-11-04741-f002:**
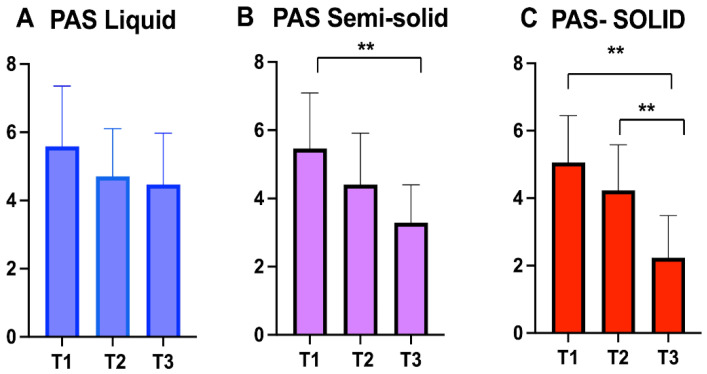
Open Partial Horizontal Laryngectomies-Penetration Aspiration Scale (OPHL-PAS) score values at three different observation point for liquids (**A**), semisolids (**B**) and solids (**C**). The asterisks represent the statistically significative difference between evaluation (** *p* < 0.001).

**Figure 3 jcm-11-04741-f003:**
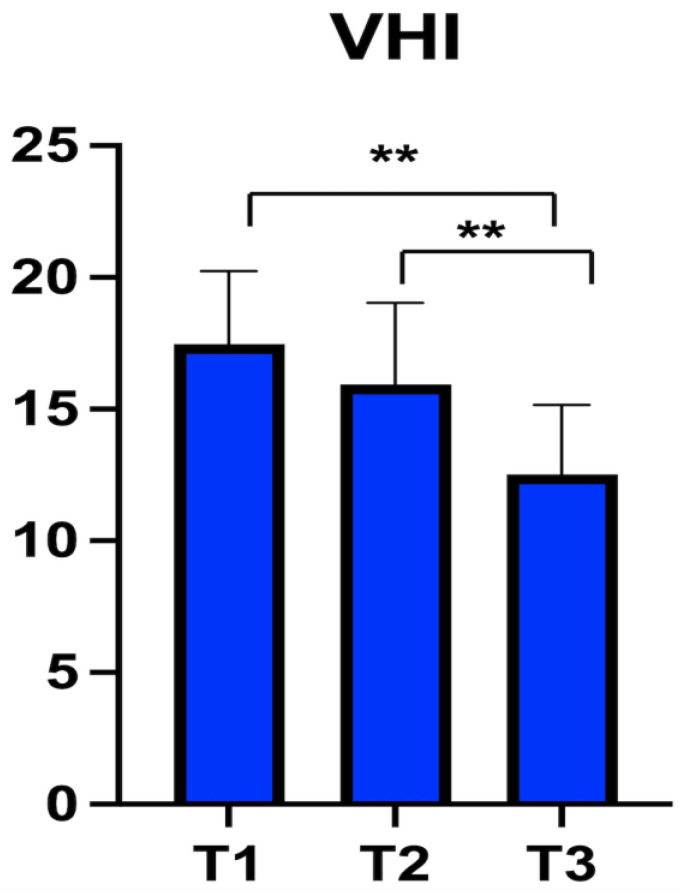
Voice Handicap Index-10 (VHI-10) score at three different observation points. The asterisks represent the statistically significative difference between evaluations (** *p* < 0.001).

**Table 1 jcm-11-04741-t001:** Demographics, surgical and oncological features of patients enrolled.

ID Case	Sex/Age	Main Symptoms	Site/Subsites	cTNM	Surgery	Surgical Time (Minutes)	pTNM	PO Therapy	FU (Months/Status)
1	M/58	Dysphonia	Glottis/RVC	cT3N0	OPHL 2B + ARY DX + RSND (II–IV)	120	pT3N2a	RT + CT	27/NED
2	M/57	Dysphonia	Glottis/RVC + AC	cT3N0	OPHL 2B + ARY DX + RSND (II–IV) + IJV	285	pT3N1	FU	26/NED
3	M/79	Dysphonia	Glottis/RVC + AC	cT3N0	OPHL 2A + ARY DX + RSND (I–IV)	180	pT4aN0	RT (rejected)	25/DOD
4	M/59	Dysphonia	Glottis/RVC	cT2N0	OPHL 2A + RSND (II–IV)	190	pT2N0	FU	22/NED
5	M/63	Dysphonia, Dysphagia	Supraglottis/infrahyoid epiglottis	cT3N0	OPHL I + BSND (I–III)	245	pT3N0	FU	21/AWD
6	M/57	Dysphonia	Glottis/LVC + AC	cT3N0	OPHL 2B + BSND (II–IV)	235	pT4aN0	RT	21/NED
7	M/74	Dysphonia	Glottis/RVC+AC	cT2N0	OPHL 2A + RSND (II–IV)	210	pT2N0	FU	19/NED
8	M/70	Dysphonia	Glottis/RVC+AC	cT3N0	OPHL 2B + ARY DX + RSND (I–IV)	140	pT3N0	FU	17/DOC
9	F/67	Dysphonia Dysphagia	Supraglottis	cT2N0	OPHL I + BSND (I–III)	195	pT3N0	FU	17/NED
10	M/61	Dysphonia	Glottis/RVC	cT2N0	OPHL 2A + RSND (II–IV)	170	pT3N0	FU	16/NED
11	M/73	Dysphonia	Glottis/LVC + AC	cT3N2a	OPHL 2B + ARY SIN + BSND I–IV	200	pT3N2b	RT + CT	15/NED
12	M/64	Dysphonia Dysphagia	Supraglottis	cT3N0	OPHL I + BSND (I–III)	210	pT3N0	FU	13/NED
13	F/68	Dysphonia	Glottis/LVC	cT2N0	OPHL 2A + ARY SN + LSND (II–IV)	170	pT2N0	RT	13/NED
14	M/79	Dysphonia	Glottis/LVC + AC	cT3N0	OPHL 2B + ARY SN + LSND (I–IV)	190	pT4aN0	RT	13/DOC
15	M/50	Dysphonia	Glottis/RVC	cT2N0	OPHL 2A + ARY DX + RSND (II–IV)	205	pT2N0	RT	11/NED
16	M/56	Dysphonia	Glottis/AC	cT2N1	OPHL 2B + BSND I–IV	230	pT2N2a	RT + CT	11/NED
17	M/68	Dysphonia	Glottis/LVC	cT3N0	OPHL 2B + ARY SIN + LSND (II–IV)	215	pT3N0	RT (perineural invasion)	9/NED

cTNM: clinical TNM; pTNM: pathological TNM; RVC: right vocal cord; AC: anterior commissure; LVC: left vocal cord; ARY: arytenoid; RSND: right selective neck dissection; IJV: internal jugular vein; LSND: left selective neck dissection; BSND: bilateral selective neck dissection; PO: post-operative; RT: radiotherapy; CT: chemotherapy; FU: follow-up; NED: no evidence of disease; DOD: died of disease; AWD: alive with disease; DOC: died for other cause.

**Table 2 jcm-11-04741-t002:** Respiration and deglutition outcomes in enrolled patients.

Case	Removal Tracheostomy Tube (Day P.O)	Start Oral Intake in Rehabilitation Protocol	Removal NGT (Day P.O)
1	6	13	56
2	19	16	183
3	13	7	(PEG)
4	6	8	67
5	8	10	24
6	5	6	43
7	9	6	38
8	12	8	27
9	7	14	34
10	5	9	47
11	-	19	31
12	9	20	36
13	8	10	39
14	35	12	54
15	10	8	38
16	37	21	67
17	18	15	58
**Average**	**12.9**	**11.9**	**52.6**
**SD**	**9.9**	**4.9**	**37.2**

P.O: post-operative; NGT: nasogastric tube; PEG: percutaneous endoscopic gastrostomy; SD: standard deviation.

**Table 3 jcm-11-04741-t003:** OPHL-PAS evaluation during the rehabilitation protocol.

Case	PAS 15 Day PO (T1)			PAS 3 Months (T2)			PAS 6 Months (T3)		
	Liquid	Semisolid	Solid	Liquid	Semisolid	Solid	Liquid	Semisolid	Solid
1	3	3	5	3	3	6	2	1	3
2	6	7	7	5	6	7	5	5	5
3	8	7	7	7	7	7	7	4	5
4	3	4	6	2	2	4	2	2	1
5	4	4	5	5	4	5	3	2	3
6	3	5	4	4	5	4	3	3	2
7	5	3	2	3	2	3	4	3	1
8	7	3	4	5	3	3	5	2	2
9	7	5	6	4	4	4	5	3	3
10	5	6	6	4	3	3	3	4	2
11	4	7	5	4	4	3	4	4	1
12	8	6	6	7	5	4	6	3	1
13	6	5	7	6	5	5	5	3	2
14	5	6	4	5	4	3	4	5	2
15	7	8	4	6	6	4	6	4	1
16	6	7	4	4	6	4	6	4	2
17	8	7	4	6	6	3	6	4	2
**Average**	**5.6**	**5.5**	**5.1**	**4.7**	**4.4**	**4.2**	**4.5**	**3.3**	**2.2**
**SD**	**1.76**	**1.62**	**1.39**	**1.4**	**1.5**	**1.34**	**1.5**	**1.1**	**1.25**

P.O: post-operative; PAS: penetration aspiration scale; SD: standard deviation.

**Table 4 jcm-11-04741-t004:** Phonation outcomes evaluated by VHI-10 of each patient.

Case	VHI-10 15 Day P.O (T1)	VHI-10 3 Months (T2)	VHI-10 6 Months (T3)
1	19	17	17
2	17	16	13
3	23	23	17
4	15	16	12
5	16	11	9
6	15	16	12
7	21	20	16
8	19	19	14
9	19	17	12
10	17	14	11
11	18	15	10
12	16	13	13
13	22	19	15
14	17	14	12
15	13	13	8
16	14	12	10
17	16	16	12
**Average**	**17.5**	**15.9**	**12.5**
**SD**	**2.8**	**3.1**	**2.6**

P.O: post-operative; VHI: voice handicap index; SD: standard deviation.

**Table 5 jcm-11-04741-t005:** Functional outcomes according to the type of OPHL.

	OPHL I	OPHL IIA	OPHL IIB
Post-operative day removal tracheostomy tube (mean ± SD)	8 ± 0.8	9.7 ± 2.4	18.6 ± 11.9
Post-operative day removal NGT (mean ± SD)	31.3 ± 5.2	45.8 ± 11.1	70.3 ± 47.2
PAS liquid (mean ± SD)	4.7 ± 1.2	4.5 ± 1.7	4.4 ± 1.4
PAS semi-solid (mean ± SD)	2.7 ± 0.5	3.3 ± 0.7	3.5 ± 1.3
PAS solid (mean ± SD)	2.3 ± 0.9	2 ± 1.4	2.6 ± 1.1
VHI-10 (mean ± SD)	11.3 ± 1.7	13.2 ± 3.1	12.5 ± 2.1

OPHL: Open Partial Horizontal Laryngectomies; NGT: nasogastric tube; PAS: Penetration Aspiration Scale; VHI: voice handicap index; SD: standard deviation.

## Data Availability

Not applicable.

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
