# Peer review of "Outcomes of Laryngeal Cancer Surgery after Open Partial Horizontal Laryngectomies with Lateral Cervical Approach"

_jcm, 2022, doi:10.3390/jcm11164741_

Round 1

Reviewer 1 Report

This article investigated postoperative swallowing, respiration, and voice functions after open partial horizontal laryngectomies. In patients who underwent OPHL, these functions are essential to maintain their QOL, therefore, this report is interesting and noteworthy. My comment is that the authors should compare the days until decanulation, the days until NGT removal, PAS score, and VHI by surgical procedures of OPHL I, IIA, and IIB. They may depend on surgical procedures.

Author Response

RESPONSE TO REVIEWERS

    Rome, August 09, 2022

Manuscript: JCM-1826970 (Outcomes of laryngeal cancer surgery after Open Partial Horizontal Laryngectomies with Lateral Cervical Approach).

Dear Reviewers,

Thanks for reviewing our manuscript and for the valuable comments that helped us clarify some relevant aspects that were missed or unclear in the first version of the paper. We have read the comments and made the changes to address comments and concerns. In addition, the manuscript has been reviewed by a native English speaker for grammar and language improvement. We hope that the changes made in the revised manuscript and responses provided below have adequately addressed the comments and made this paper stronger.

REVIEWER #1

This article investigated postoperative swallowing, respiration, and voice functions after open partial horizontal laryngectomies. In patients who underwent OPHL, these functions are essential to maintain their QOL, therefore, this report is interesting and noteworthy. My comment is that the authors should compare the days until decanulation, the days until NGT removal, PAS score, and VHI by surgical procedures of OPHL I, IIA, and IIB. They may depend on surgical procedures.

Thanks for your careful review of this study and we appreciate the comment. We added a new table (table 5) where we compare the functional outcomes according the surgical procedure.

We really appreciated your careful and thoughtful evaluation of our manuscript and hope that this revised version meets with your approval. We have tracked all changes using the “track changes” tool of Microsoft Word.

Thanks again for your interest in our work. We await your review of our revised manuscript.

Sincerely yours,

The Authors

Reviewer 2 Report

Very interesting and decent work.

I only one suggestion:

Is that possible to include either MRI or CT scans before and after the surgeries at least for one patient. So that audience can have a better idea about the surgery effects to the overall larynx.

Author Response

RESPONSE TO REVIEWERS

    Rome, August 09, 2022

Manuscript: JCM-1826970 (Outcomes of laryngeal cancer surgery after Open Partial Horizontal Laryngectomies with Lateral Cervical Approach).

Dear Reviewers,

Thanks for reviewing our manuscript and for the valuable comments that helped us clarify some relevant aspects that were missed or unclear in the first version of the paper. We have read the comments and made the changes to address comments and concerns. In addition, the manuscript has been reviewed by a native English speaker for grammar and language improvement. We hope that the changes made in the revised manuscript and responses provided below have adequately addressed the comments and made this paper stronger.

REVIEWER #2

Very interesting and decent work.

I only one suggestion:

Is that possible to include either MRI or CT scans before and after the surgeries at least for one patient. So that audience can have a better idea about the surgery effects to the overall larynx.

We shared this comment. We added in the main document two CT images before and after larynx surgery (figure 1A and 1B). We also delete the previous figure 1 because we do not have the copyright permission for the figure.

We really appreciated your careful and thoughtful evaluation of our manuscript and hope that this revised version meets with your approval. We have tracked all changes using the “track changes” tool of Microsoft Word.

Thanks again for your interest in our work. We await your review of our revised manuscript.

Sincerely yours,

The Authors

Reviewer 3 Report

The manuscript is well-designed and adds valuable information to the literature regarding OPHL and LCA treatment outcomes.  In my opinion, the findings of this work are interesting, but before being published in JCM some adjustments need to be made: 

1. The manuscript would benefit from moderate English editing for sure.  

2. Abstract - the results part is difficult to read. Try to be more specific. For example, the variation of VHI-10 was significantly different. What does it mean? Was it statistically improved over time? 

3. Introduction - I would like to see some more information about LCA approach here. Two-three sentences. 

4. Materials and Methods - Please provide the number of the Ethical Comittee Aproval.  

5. Please check the spelling of anatomical structures. Like line 87 - "homohyoid muscle". Do you mean omohyoid muscle? 

6. Results - line 167. "16 were decannulated at the end of follow up". Do you mean at hospital discharge? 

7. Results - line 174. "started 11.9 ± 4.9 days after surgery". What is ± value? SD? Please make it clear in M&M? 

8. Results - line 220-222. There was a statistically significant difference between T1 and T3. Was the difference for better or worse? Please try to specify.  Same for lines 273-274. 

9. Discussion - line 301-302. Please double check. The numbers are not right. 5/38 is not 87%. 

10. Discussion - line 305. Do you mean "Time to decannulation"? 

Once again many issues can be fixed with English editing. 

Author Response

RESPONSE TO REVIEWERS

    Rome, August 09, 2022

Manuscript: JCM-1826970 (Outcomes of laryngeal cancer surgery after Open Partial Horizontal Laryngectomies with Lateral Cervical Approach).

Dear Reviewers,

Thanks for reviewing our manuscript and for the valuable comments that helped us clarify some relevant aspects that were missed or unclear in the first version of the paper. We have read the comments and made the changes to address comments and concerns. In addition, the manuscript has been reviewed by a native English speaker for grammar and language improvement. We hope that the changes made in the revised manuscript and responses provided below have adequately addressed the comments and made this paper stronger.

REVIEWER #3

The manuscript is well-designed and adds valuable information to the literature regarding OPHL and LCA treatment outcomes. In my opinion, the findings of this work are interesting, but before being published in JCM some adjustments need to be made.

  1. The manuscript would benefit from moderate English editing for sure.

Thanks for this comment. The paper has been carefully edited for grammar and language errors from an English native speaker.

  1. Abstract - the results part is difficult to read. Try to be more specific. For example, the variation of VHI-10 was significantly different. What does it mean? Was it statistically improved over time?

Thanks for this comment and we shared it. We change the result section of the abstract as following: “The functional outcomes of the LCA are stackable with that of the classical anterior cervical approach in terms of respiration, swallowing and speech. One-way ANOVA was performed to evaluate the variances of PAS and VHI scores at the three different observation points. No statistically significant differences were observed in the OPHL- PAS score for liquid (p=0.1) at the three different observation point. Statistically significant improvement was observed in OPHL- PAS score for semisolid and solid (p<0.00001) between T1 and T3 (p=0.0001) and for solid between T2 and T3 (p<0.00001). The improvement of VHI-10 was statistically significative (p<0.00001) between T1 and T2 the three different observation points (T1-T2 and T2-T3).”

  1. Introduction - I would like to see some more information about LCA approach here. Two-three sentences.

Thanks for this comment. We added, in the introduction section, the following sentences. “This approach is performed by a skin incision following the anterior border of the sternocleidomastoid muscle, the preparation of an anterior myocutaneous (AMC) in order to create a space of work to perform the lateral neck dissection, the partial laryngectomy and the pexy.”

  1. Materials and Methods - Please provide the number of the Ethical Comittee Aproval.

Thank for this comment. We performed a type of laryngeal surgery described in literature and approved all over the world. So in our opinion the Ethics Committee or Institutional Review Board approval was not necessary.

  1. Please check the spelling of anatomical structures. Like line 87 - "homohyoid muscle". Do you mean omohyoid muscle?

Thank for this comment. We change “homohyoid muscle” in “omohyoid muscle” in the text

  1. Results - line 167. "16 were decannulated at the end of follow up". Do you mean at hospital discharge?

Thank for this comment. We mean at the end of the time of this study. So we change the words “ at the end of follow up” in “ at the end of six months of observation of this study”

  1. Results - line 174. "started 11.9 ± 4.9 days after surgery". What is ± value? SD? Please make it clear in M&M?

Thank for this comment. With ± we mean the standard deviation of the mean value. For this reason, we added in the Materials and Methods – Statistical analysis section the following sentence: “Data are indicated as mean ± standard deviation, range and percentage.”

  1. Results - line 220-222. There was a statistically significant difference between T1 and T3. Was the difference for better or worse? Please try to specify. Same for lines 273-274.

Thank for this comment. We specify the results outcomes for swallowing and for phonation.

In particular the change swallowing section of the results in: “FEES evaluation was repeated 15 days, 3 and 6 months after surgery (table 3). No statistically significant improvement were observed in the OPHL-PAS score for liquid (ANOVA: p=0.1) at the three different observation point (Figure 3A). For OPHL-PAS score for semisolid bolus a statistically significant improvement was observed in OPHL-PAS score for semisolid over the six months of observation (ANOVA: p=0.0003); in particular a significant difference was observed between T1 and T3 (BH: p=0.0001) (Figure 3B). Statistically significant difference was observed OPHL-PAS score for solid over the period of the study (ANOVA: p< 0.00001); in particular a significant difference improvement was observed between T1 and T3 (BH: p=0.0001) and T2 and T3 (HB: p< 0.00001) (Figure 3C).

In the phonation section we change the sentences as following: “ The variation of VHI-10 between the three different observation points was statistically significant (ANOVA: p< 0.00001). The statistically significant improvement was observed between T1 and T3 (BH: p <0.00001) and T2 and T3 (BH: p=0.002) (Figure 4).”

  1. Discussion - line 301-302. Please double check. The numbers are not right. 5/38 is not 87%.

Thank for this comment. We change 5/38 in 33/38.

  1. Discussion - line 305. Do you mean "Time to decannulation"?

Thank for the comment. We mean the time among the surgery and the post-operative day when the tracheostomy is closed.

  1. Once again many issues can be fixed with English editing.

After the revision performed, the paper has been carefully edited for grammar and language errors from an English native speaker.

We really appreciated your careful and thoughtful evaluation of our manuscript and hope that this revised version meets with your approval. We have tracked all changes using the “track changes” tool of Microsoft Word.

Thanks again for your interest in our work. We await your review of our revised manuscript.

Sincerely yours,

The Authors